# Utilisation of far infrared-emitting garments for optimising performance and recovery in sport: Real potential or new fad? A systematic review

**Bastien Bontemps**[1,2], **Mathieu Gruet**[1], **Fabrice Vercruyssen**[1], **Julien Louis**[2]*

**1** Université de Toulon, Laboratoire IAPS, Toulon, France, **2** Research Institute for Sport and Exercise Sciences, Liverpool John Moores University, Liverpool, United Kingdom

* J.B.Louis@ljmu.ac.uk

## Abstract

### Background

Thanks to the specific materials they embed, far infrared (FIR)-emitting garments can interact with the body's physiological functions. Such effects have been sought in medicine and physiotherapy for a long time for the treatment/relief of a variety of pathologies and disabling conditions. Recently, FIR-emitting garments have been introduced in the sporting domain under the influence of manufacturers seeing here a new opportunity to support physical performance in athletes, though this is not clearly established. To fill this gap, in this systematic review, we summarize the scientific evidence on the use of FIR-emitting garments in sport and provide directions for future research by shedding light on current scientific limitations.

### Method

Five scientific databases (PubMed, Cochrane, ScienceDirect, Scopus and SPORTDiscus) were searched by two independent reviewers. Studies investigating the effects of FIR-emitting garments on at least one physiological outcome related to exercise performance and/or recovery in humans were selected. The methodological quality of retained studies was assessed using the Physiotherapy Evidence Database (PEDro) scale.

### Results and discussion

Eleven studies met the inclusion criteria and were included in the systematic review. Studies investigating similar outcomes related to exercise performance or recovery were scarce and results inconclusive, which prevents from drawing firm conclusion about the utilisation of FIR-emitting garments in athletes. However, these early results show that FIR-emitting garments may be of interest for exercise performance and recovery, mainly through their effects on the body's thermoregulation and haemodynamic function. The summary provided in this review can be used to inform the design of future studies. (PROSPERO registration number: CRD42021238029).

**Data Availability Statement:** All relevant data are within the paper and its Supporting information files.

**Funding:** The authors received no specific funding for this work.

**Competing interests:** The authors have declared that no competing interests exist.

## Introduction

In the ever-developing area of sports performance, sportswear manufacturers now seat alongside coaches and athletes in their quest for performance optimisation; the 2019 sub-2-h attempt on the marathon by Eliud Kipchogue being one of the most recent examples. Sportswear development has been fostered by advances in nanotechnology and touted by manufacturers as new means of enhancing performance and/or optimizing the recovery process in athletes. For example, compression garments have garnered wide acceptance among athletes for utilisation during exercise and/or recovery for about 15 years. Initially developed for medical purposes, compression garments are now considered by manufacturers as a benchmark model of technological innovation. Far infrared (FIR)-emitting garments have recently been introduced in the sporting area with the potential to improve exercise performance and recovery. FIR-emitting garments are more and more used by athletes, including elite athletes, as shown in the 2020 attempt to break the 24-h world record in running, though their effects on exercise performance and/or recovery are not clearly established.

FIR is a specific band (50 to 1000 μm) in the infrared (IR) spectrum of electromagnetic radiation [1]. FIR radiations are emitted by any object presenting a temperature above the absolute zero, including the human body at room temperature. While FIR is undetectable by human eyesight, skin thermoreceptors can perceive FIR as radiant heat. Interestingly, thanks to its high water content, the human body can absorb FIR, which may penetrate up to 4 cm into human tissues [2]. FIR may, therefore, excite molecules and cells (i.e. cytochrome-c-oxidase and intracellular water) and alter biological functions [2]. FIR have been used for a long time in medicine and physiotherapy for the treatment/relief of a variety of pathologies and disabling conditions (e.g. chronic muscle and joint pain and fatigue, wound healing, heart failure, disturbed sleep, lactation issue, dysmenorrhea). For a summary of the scientific literature on medical applications of FIR, the reader is referred to Tsai and Hamblin [2]. To date, our understanding of the biological underpinning of FIR therapy mainly derives from *in vitro* studies or studies involving animal models. In these experimental conditions, FIR-emitting materials would confer anti-inflammatory [3], antioxidant [4] and antimicrobial [5] effects as well as rejuvenating skin and potentially muscle-tendon unit tissue [6]. These effects may delay the appearance of muscle fatigue [7], promote peripheral microcirculation [8] and cerebral blood flow [9] and enhance blood circulation and metabolism [10].

Athletes may use a variety of FIR techniques that include FIR cabins/saunas, FIR ray devices (i.e. IR heat lamp, tourmaline/jade heating pad, heating bag) and recent FIR-emitting garments [2]. These innovative garments have been developed as a practical and flexible alternative to traditional FIR therapies that could be implemented only prior to or post-exercise [11]. FIR-emitting garments are generally composed of bioceramic fibres or powders embedded in the fabric, or bioceramic disks/patches positioned onto the fabric. Bioceramic materials are composed of various metallic oxides (e.g. magnesium, silica, aluminium, tourmaline), that can radiate and/or provide IR and FIR when heated to body temperature [12]. As such, it has been suggested that FIR-emitting garments can absorb IR/heat energy emitted by the body, maintain their temperature, and re-emit IR into radiation within the FIR wavelength back to the body, through convection and conduction mechanisms. Whilst FIR-emitting garments confer no thermal heating effect at room temperature [13], they could represent an interesting means of delivering localised biological effects to the human body. To date, the use of FIR-emitting garments at rest has been shown to enhance peripheral blood flow [14], muscle tissue oxygenation [14,15], resting metabolic rate [16] and subjective sleep parameters [17]. Although the use of FIR-emitting garments is spreading among athletes for different purposes including optimisation of warm-up, exercise performance, and post-exercise recovery, their effectiveness is not clear.

Within this context, the purpose of this systematic review is to provide a synthesis of current evidence on the potential effectiveness of FIR-emitting garments in improving physical performance and/or recovery in healthy adults.

## Methods

This systematic review is presented using the Preferred Method Items for Systematic Reviews and Meta-Analysis (PRISMA) statement format [18]. This review has been registered in PROSPERO (registration number: CRD42021238029).

### Literature search

An electronic literature search was conducted in November 2020 in accordance with PRISMA guidelines for systematic reviews (Fig 1). Studies were acquired by searching (by two reviewers, B.B. and L.J.) the electronic database of the US National Library of Medicine (PubMed), Cochrane, ScienceDirect, Scopus and SPORTDiscus using the following combination of key-words: ((("*bioceramic*" OR "*ceramic*" OR "*far infrared*" OR "*FIR*") AND (*garment* OR *clothing* OR *apparel* OR "*fabric*" OR "*glove*" OR "*sock*") AND (*exercise* OR *performance* OR *recovery* OR *physical activity* OR *physical exertion* OR *fatigue* OR "*muscle contraction*" OR "*muscle damage*"))). MeSH terms were used if the databases permitted it. Original articles published in English or French language only were kept. Once the initial search was completed, duplicates were removed. Titles and abstracts of remaining studies were screened for eligibility. The remaining studies were read in full and assessed for eligibility and included in the final review and analysis. Electronic database searching was supplemented by examining reference sections of relevant articles (i.e. hand search).

### Selection criteria/study eligibility

Exclusion/inclusion criteria were identified using the acronym "PICOS": population (i.e. healthy adults > 18 years of age), intervention (i.e. FIR-emitting garments worn pre-, during or post-exercise), comparator (i.e. placebo and/or control groups), outcomes (i.e. all physiological or perceptive parameters related to exercise performance and/or recovery) and study type (i.e. all experimental studies). Studies investigating any other FIR-emitting therapy such as FIR saunas or electrically powered devices were excluded due to their greater irradiance and/or power density. Textile engineering studies not conducted in the domain of sport or exercise performance were excluded from the analysis. We did not consider studies using *in vivo* and/or *in vitro* animal models. No limit to the search domain was applied regarding the training level of participants, exercise modalities and characteristics, intervention designs, measured physiological and perceptive outcomes, or FIR-emitting garments designs (i.e. composition, emissivity, tightness, or body area covered). Considering the low number of eligible studies, and the large variability in populations, interventions, and the selected variables to report the outcomes between the different studies, it was deemed inappropriate to complement this systematic review with a meta-analysis [19].

### Quality assessment

The methodological quality of studies was assessed using the Physiotherapy Evidence Database (PEDro) scale [20]. The 11-point PEDro scale has been shown to be an acceptable method of reliably assessing the internal validity of randomised control trials. Items include randomisation and allocation, blinding, selective reporting, and statistical analysis criteria. The first item point is awarded for stating eligibility criteria and is not summed in the final scores. Studies

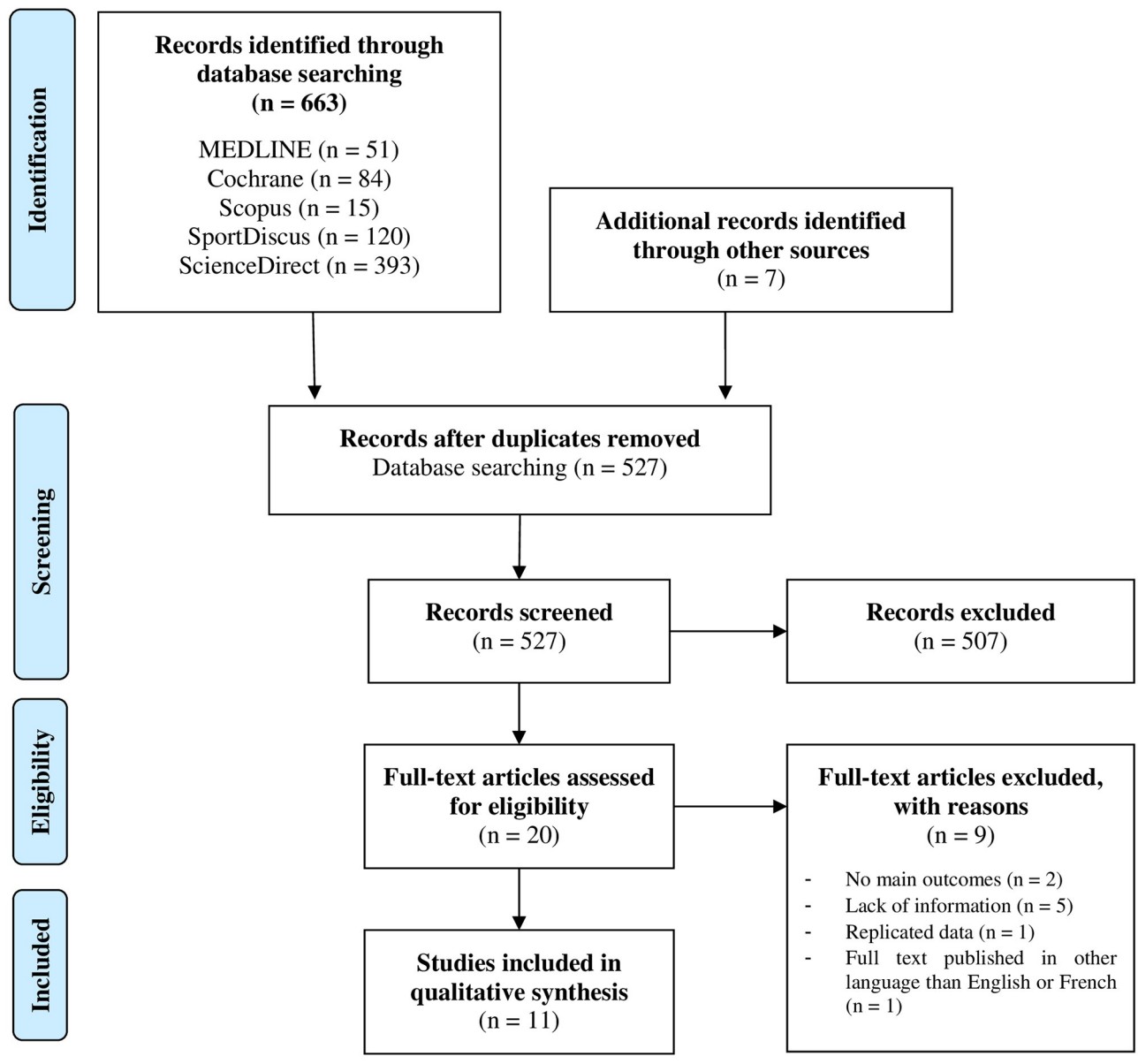

**Fig 1. Process of elimination and inclusion of studies for review based on PRISMA guidelines.**

are classified in a range from 0 to 10 points with a sum score of 0 to 3 rated as poor, 4 to 6 as fair, 7 to 8 as good and 9 to 10 as excellent quality. We did not consider PEDro score as an exclusion criterion in this review. Identification of PEDro scores were carried out by B.B. and L.J. independently, and a consensus was reached through scores comparison and deliberation when disagreements were found.

## Results

### Search results and study overview

The literature search of the different databases resulted in 663 records. After duplicates were removed, a total of 520 records were still potentially eligible. An additional seven studies were

identified and supplemented by examining reference sections of relevant articles, yielding a total of 527 records. Remaining studies were screened using titles and abstracts for an acceptable interventional protocol investigating the use of FIR-emitting garments and evaluating their potential benefits on at least one physiological and/or perceptive parameter related to exercise performance or recovery. It resulted in a selection of twenty studies which were then read in full to ensure they met the inclusion and exclusion criteria. Of the studies that were read in full, two studies were excluded because they did not present the main outcomes [21,22], one study was excluded for replicated data [23], five studies were excluded for lack of information [24–28], and one study was excluded because it was published in another language than English or French, although the abstract was written in English [29]. A total of eleven studies were ultimately identified and included in this review (Table 1). Thereafter, selected studies were classified into two different categories according to their experimental design: six studies investigated the '*effects of FIR-emitting garments worn during exercise*' [16,30–34], and five studies investigated the '*effects of FIR-emitting garments worn post-exercise*' [35–39].

## Samples

Most of the studies testing FIR-emitting garments during exercise recruited healthy participants [16,30,32,34] or recreationally trained participants [31,33], although one study included well trained participants [32]. Participants' age ranged from 18 to 57 years. All the studies had samples sizes of five to twenty participants, with only males [30,31,34] or a mixture of female and male participants [16,32,33]. In the studies testing FIR-emitting garments post-exercise, participants were recreationally active [36–38] or elite male athletes [35,39], and their age ranged from 19 to 26 years except in one study where the participants were > 60 years old [37]. Sample sizes ranged from 14 to 22 participants.

## Exercise modalities

Nine of the eleven studies used an acute exercise modality [16,30–35,37,38], and two studies used consecutive bouts of exercise (i.e. two sessions per day separated by 3 to 4 hours over one day to 2 weeks) [36,39]. The exercise protocols varied across studies, including whole-body plyometric (i.e. 100 consecutive drop-jumps) [35] and isokinetic eccentric (i.e. $3 \times 30$ maximal eccentric contractions of knee extensors) exercise bouts [38], incremental cycling exercises [31,33], ecological running time trial (i.e. 10-km time trial) [34], endurance laboratory tests (i.e. uphill walking) [37], postural test (i.e. standard standing postural task and handstand stabilization) [32] and intense training over consecutive days (i.e. simulation of a typical training day in futsal athletes) [39].

## Far infrared-emitting garment solutions

Studies used a variety of FIR-emitting garments. In ten studies, FIR-emitting garments were applied on the lower body (i.e. feet, thigh, calves or whole leg covered) [30–39], in seven studies it was on the upper body (i.e. arm, forearm, trunk or all covered simultaneously) [16,30–34,36], and in six studies garments covered both the lower and upper body simultaneously [30–34,36]. In studies testing the garments during exercise, the garments were applied on the lower and upper body [30–34] except in one study using only a FIR-emitting t-shirt [16]. In contrast, in studies testing the garments post-exercise, only the lower limbs were covered [35,37–39] except in one study using the garments on both the lower and upper body [36]. Garments were t-shirts (4 studies; [16,30,33,34]), long sleeved shirts (3 studies; [31,32,36]), shorts (2 studies; [33,34]), trousers (7 studies; [30–32,35,36,38,39]) and shin guards or

**Table 1. Summary of studies included in the review examining the use of innovative FIR-emitting garments during physical exercise and during the post-exercise recovery period.**

| Reference | Participants | Study design | FIR-emitting intervention | Protocol | Main outcomes |
|---|---|---|---|---|---|
| **DURING EXERCISE** | | | | | |
| **Katsuura et al.** [30] | 7 healthy male students | Cross-over, control-innovative garments trial | Bioceramic t-shirts and trousers | 30 min cycling at 75 W | + Forearm blood flow at 20 and 30 min<br>↔ Skin temperature<br>↔ Oxygen uptake<br>↔ Cardiac output (higher from 10 to 20 min only)<br>↔ HR (lower from 20 to 30 min only) |
| **Leung et al.** [16] | Experiment 2: 5 non-athletes | Double-blind, cross-over, control-innovative garments trial | Bioceramic t-shirt | Experiment 2: 30 min TRT with 2% slope | ↔ Tiredness<br>↔ Skin temperature<br>↔ Stabilization of respiration and heart rate (i.e. reflecting parasympathetic control) |
| **Worobets et al.** [31] | 12 recreational male adult cyclists | Cross-over, placebo-controlled trial | Bioceramic long sleeved shirts and trousers | Incremental cycling test (+25 W every 2 min), 2 times per condition, R' = at least 48 h | + Oxygen uptake at low intensity (blood lactate ≤ 4 mmol/L)<br>↔ Oxygen uptake at high intensity (blood lactate > 4 mmol/L) |
| **Cian et al.** [32] | Experiment 1: 12 healthy participants (6 females and 6 males)<br>Experiment 2: 14 well trained adult gymnasts (5 females and 9 males) | Cross-over, double-blind, placebo-controlled trial | Bioceramic long suits (upper + lower body) | Experiment 1: standard standing postural task on a force platform (4 × 60s per condition with r' = 1 min; R' = 10 min)<br>Experiment 2: Handstand stabilization task (5 × 5-s per condition with r' = 1-min; R' = 10-min) | + Postural stability<br>+ Postural control |
| **Mantegazza et al.** [33] | 20 recreationally active adults (9 females and 11 males) | Cross-over, double-blind, placebo-controlled trial | Bioceramic t-shirt and shorts | Maximal and incremental cardiopulmonary test on a cycle ergometer | + Maximal exercise capacity (i.e. endurance time and $\dot{V}O_{2peak}$)<br>+ Delayed anaerobic threshold<br>+ Blood lactate at 10-min exercise<br>↔ Ventilation at $\dot{V}O_{2peak}$<br>↔ No influence on participants' baseline characteristics |
| **Furlan et al.** [34] | 10 healthy young males | Cross-over, single-blind (except for control), placebo and control-innovative garments trial | Bioceramic t-shirt, short and shin guards | 10 km running performance, R' = 3 to 7 days | ↔ Total performance time (ES = -0.23 compared to control)<br>↔ RPE (ES = 0.25 compared to control)<br>↔ DOMS (ES = 0.47 compared to control)<br>↔ Blood lactate<br>↔ HR |
| **DURING POST-EXERCISE RECOVERY** | | | | | |
| **Loturco et al.** [35] | 21 elite young male soccer players | Double-blind, placebo-controlled trial | Bioceramic trousers worn 6-h post exercise through a 10-h sleeping period for 3 consecutive days | 100 consecutive drop-jumps (as high as possible above a 45-cm height box, r' = 6-s) | ↔ DOMS (ES = 0.73 and 0.84 at 48h and 72h post exercise)<br>↔ Mid-thigh circumference<br>↔ Plasma CK elevation<br>↔ Vertical jumps (i.e. CMJ, SJ)<br>↔ Maximal dynamic strength (i.e. leg press 1 RM) |

(*Continued*)

**Table 1.** (Continued)

| Reference | Participants | Study design | FIR-emitting intervention | Protocol | Main outcomes |
|---|---|---|---|---|---|
| **Zinke et al.** [36] | 14 resistance trained male adults | Cross-over, single-blind, placebo-controlled trial | Platinum harmonized long sleeved shirts and trousers worn between training sessions for 3h and during sleep periods for 3 subsequent nights | Simulation of a typical training day in athletes (2 sessions in one day, R' = 3-h)<br>Power training in the morning (i.e. unilateral alternating step-up jumps and CMJ)<br>High volume of strength exercise in the afternoon (i.e. squats) | ↔ Force decrements post-exercise (i.e. MVC)<br>↔ Peripheral (i.e. potentiated twitches) and central (i.e. voluntary activation) component of fatigue<br>↔ Jump performance<br>↔ Plasma CK activity<br>↔ Sleep duration (ES = 0.8)<br>↔ Perceived intensity (ES = 0.8 at 24h post training sessions)<br>↔ Perceived recovery (ES = 0.6 and 0.7 at 24h and 48h post training sessions) |
| **Black et al.** [37] | 16 healthy older adults | Cross-over, double-blind, control-innovative garments trial | Bioceramic stockings worn during subsequent days including testing blocks; not worn during sleep time and water-based activities | Uphill treadmill walking (35 min in total at ~ 4.4 km.h$^{-1}$; 4 × 5 min at +10% slope; r' = 2-min active recovery at 0% slope; 4-min cool-down at 0% slope) | ↔ Postural balance (very likely effect at 36h post exercise)<br>↔ Resting microvascular perfusion (possible to likely effects on muscle microvascular blood flow and oxygen consumption at rest from 14 to 62h post exercise; Possible effect on muscle oxygen consumption during plantar flexion exercise at 24h post exercise)<br>↔ Range of motion (Possible to likely effects on ankle dorsiflexion and plantar flexion at 38h and 62h post exercise)<br>↔ Oxygen consumption |
| **Nunes et al.** [38] | 22 moderately active males | Double-blind, placebo-controlled trial | Bioceramic trousers worn 2h post exercise and 2h before subsequent evaluations | 3 × 30 maximal ECC of KE (60°.s$^{-1}$, r' = 30s) | ↔ Force decrements post-exercise<br>↔ Plasma CK elevation (ES = 0.50 and -0.58 at 24-h and 48h postexercise compared to placebo)<br>↔ Lactate dehydrogenase<br>↔ DOMS<br>↔ Perceived recovery status |
| **Nunes et al.** [39] | 20 elite futsal players | Double-blind, placebo-controlled trial | Bioceramic trousers worn during the sleep time for 5 consecutive days throughout a 2-week training camp | Preseason training program (2 sessions/day, 90 to 120-min per session) | ↔ 5-metre sprints in week 2<br>↔ Train strain in week 1<br>↔ SJ in week 2 (ES = 0.38) and CMJ in week 1 (ES = 0.26)<br>↔ 10-metre sprints in week 1 (ES = -0.35) and 15-metre sprints<br>↔ Moderate to large effect on DOMS in week 1, but unclear in week 2<br>↔ TNF-a (ES = 0.21 and 0.34 in week 1 and 2, respectively) and IL-10 (ES = 0.48 and 0.45 in week 1 and 2, respectively)<br>↔ Oxidative stress |

CK: Creatine kinase; DOMS: Delayed onset muscle soreness; HR: Heart rate; HST: Harvard Step Test; R': Recovery time between series or conditions; r': Recovery time between sets; TRT: Treadmill running test; $\dot{V}O_{2peak}$: Peak oxygen uptake; +: Statistically significant difference between garment conditions (p < 0.05); ↔: No statistical difference between garment conditions (p > 0.05); ES: Cohen's *d* effect size between garment conditions.

stockings (2 studies; [34,37]). Garments tightness was not clearly described in six studies [16,30,33,34,38,39], loose in four studies [31,35,36] and skin-fit (non-compressive) in two studies [32,37]. The metallic oxides composing the FIR garments and/or emissivity characteristics were both clearly described in only four studies [16,35,38,39]. Exposure duration to FIR-emitting garments was short (< 90 min) and acute (only one session) in all the exercise protocols [16,30–34]. On the contrary, exposure duration was varied (from 120 min to 10 h) and chronic (up to 5 consecutive days) in the post-exercise recovery protocols [35–39].

## PEDro scores of included studies

The mean PEDro scores across all studies are 6.2 and 8.0 for the 'studies testing FIR-emitting garments during exercise' and the 'studies testing FIR-emitting garments post-exercise', respectively. Four studies had a PEDro score ≤ 6 which is considered rather low [30,31,34,36], while seven studies had a PEDro score ≥ 7 which is regarded as high quality [16,32,33,35,37–39] with four studies scoring 9 [33,37–39] and two studies scoring 8 [16,35] (Table 2). The individual PEDro items that were the least achieved were randomisation [16,30–32,34,36], concealed allocation [30–32,34,36], report of the number of subjects from whom key outcomes were measured [35–39] and data measured at baseline prior to intervention [16,31,32,34].

## Overview of the main findings

An overview of the main outcomes from selected studies is displayed in Table 1. Overall, studies investigating similar outcomes are scarce and results inconclusive. The relevant findings are further presented below according to their main outcomes.

   **Studies testing FIR-emitting garments during exercise.**   The effect of FIR-emitting garments on oxygen uptake ($\dot{V}O_2$) was evaluated in three studies [30,31,33]. Worobets et al. [31] reported a significant decrease in $\dot{V}O_2$ (-1%) in recreational cyclists exercising at low (< 2 mmol.L$^{-1}$ of blood lactate; p = 0.014) and moderate (< 4 mmol.L$^{-1}$ of blood lactate; p = 0.048) intensities while wearing FIR-emitting long sleeved shirts and trousers compared to control garments. In contrast, Katsuura et al. [30] did not report any significant change (p > 0.05) in $\dot{V}O_2$ during a 30-min cycling exercise at very low intensity (75 W) despite forearm blood flow being greater at 20 and 30 min using FIR-emitting garments compared to control garments. At higher intensity (4 to 6 mmol.L$^{-1}$ of blood lactate), $\dot{V}O_2$ did not change either in recreational cyclists wearing FIR-emitting garments (p = 0.511) [31]. However, Mantegazza et al. [33] recently reported a higher $\dot{V}O_{2peak}$ value (+5%; p = 0.006), endurance time (+4%;

**Table 2. PEDro scores for the selected studies.**

| Study | Item 1 | Item 2 | Item 3 | Item 4 | Item 5 | Item 6 | Item 7 | Item 8 | Item 9 | Item 10 | Item 11 | Total score |
|---|---|---|---|---|---|---|---|---|---|---|---|---|
| **Katsuura et al. [30]** | 0 | 0 | 0 | 1 | 0 | 0 | 0 | 1 | 1 | 1 | 1 | **5** |
| **Leung et al. [16]** | 0 | 0 | 1 | 0 | 1 | 1 | 1 | 1 | 1 | 1 | 1 | **8** |
| **Worobets et al [31]** | 0 | 0 | 0 | 0 | 0 | 0 | 0 | 1 | 1 | 1 | 0 | **3** |
| **Cian et al. [32]** | 1 | 0 | 0 | 0 | 1 | 1 | 1 | 1 | 1 | 1 | 1 | **7** |
| **Mantegazza et al. [33]** | 0 | 1 | 1 | 0 | 1 | 1 | 1 | 1 | 1 | 1 | 1 | **9** |
| **Furlan et al. [34]** | 1 | 0 | 0 | 0 | 1 | 0 | 0 | 1 | 1 | 1 | 1 | **5** |
| **Loturco et al. [35]** | 1 | 0 | 1 | 1 | 1 | 1 | 1 | 0 | 1 | 1 | 1 | **8** |
| **Zinke et al. [36]** | 1 | 0 | 0 | 1 | 1 | 0 | 0 | 0 | 1 | 1 | 1 | **5** |
| **Black et al. [37]** | 1 | 1 | 1 | 1 | 1 | 1 | 1 | 0 | 1 | 1 | 1 | **9** |
| **Nunes et al. [38]** | 1 | 1 | 1 | 1 | 1 | 1 | 1 | 0 | 1 | 1 | 1 | **9** |
| **Nunes et al. [39]** | 1 | 1 | 1 | 1 | 1 | 1 | 1 | 0 | 1 | 1 | 1 | **9** |

p = 0.009), and $\dot{V}O_2$ at the anaerobic threshold (+13%; p = 0.005) during an incremental test in sedentary participants wearing FIR-emitting garments compared to placebo garments.

Two studies investigated skin temperature measured during exercise using FIR-emitting garments [16,30]. No significant difference between garments were observed (p > 0.05), whatever the time point during the exercise [16,30]. Although a tendency towards a lowered skin temperature was recorded over the exercise period with the FIR-emitting garments compared to the control garments, no effect sizes were reported which limits the interpretation of these results.

Only one study evaluated the effects of FIR-emitting garments on performance in an ecological (i.e. 10-km on a running track) time-trial in recreational runners [34]. No significant difference in completion time was observed between experimental conditions (i.e. control, placebo and FIR-emitting garments; p = 0.06). Only a small effect size (ES = -0.23) was reported for completion time between FIR-emitting and control garments conditions. Interestingly, no difference was observed for heart rate (p = 0.80), perceived exertion (p = 0.80), delayed onset muscle soreness (DOMS) and peak blood lactate (p = 0.802) between garment conditions. In line with these results, Loturco et al. [16] did not report significant changes in perceptual responses and heart rate during a 30-min treadmill walking test in untrained participants wearing a FIR-emitting shirt compared to a control shirt.

The use of FIR-emitting garments (two-piece black suits covering the whole-body) was also tested during standard standing postural tests in untrained males, and during a handstand stabilization task in expert male gymnasts [32]. Authors reported significant improvements in postural stability and control in both groups of participants thanks to the FIR-emitting garments compared to placebo garments.

**Studies testing FIR-emitting garments post-exercise.** The effect of FIR-emitting garments worn exclusively overnight was evaluated in three studies [35,36,39]. FIR-emitting trousers were tested during three consecutive nights (10h per night) in young football players following a plyometric exercise bout (100 consecutive maximal 45-cm drop-jumps) [35]. Authors did not report a significant effect (p > 0.05) on the recovery kinetic of maximal isometric and dynamic (i.e. counter movement jump, squat jump and 1RM leg press) muscle strength, thigh circumferences and blood creatine kinase activity (CK). Although no significant difference between conditions was observed, Cohen's effect sizes were moderate (ES = 0.73) and large (ES = 0.84) for DOMS in favour of FIR-emitting garments compared to placebo garments at 48h and 72h post-exercise, respectively [35]. When FIR-emitting garments were worn overnight (5 consecutive nights per week) over a 2-week preseason training camp in elite futsal players, small positive effect sizes were observed for the change in squat jump performance at week 2 (ES = 0.38), countermovement jump performance at week 1 (ES = 0.26), 10-metre sprint (ES = 0.35) as well as training strain (ES = -1.19; p < 0.05) at week 1, with FIR-emitting garments compared to placebo garments [39]. Regarding indirect markers of muscle damage, authors reported moderate and large positive effect sizes for DOMS in seven training sessions accompanied with likely small effects (ES = 0.48 and 0.45 at week 1 and 2) on IL-10 and oxidative stress (ES = 1.25; p < 0.05) at week 1 [39]. Zinke et al. [36] also revealed that wearing FIR-emitting garments during the three consecutive nights following a simulation of a typical training day in resistance-trained athletes improved recovery by lowering muscle pain intensity (ES = 0.8 and 0.6 at 24h and 48h post-training day) and improving perceived recovery (ES = 0.6 and 0.7 at 24-h and 48-h post-training day), and sleep quality on the third night (ES = 0.8).

Three studies evaluated the effects of FIR-emitting garments worn immediately post-exercise and/or between bouts of exercise on the recovery process [36–38]. Nunes et al. [38] did

not report significant benefits (p > 0.05) on maximal voluntary contraction (MVC), blood CK, knee extensors (KE) DOMS or perceived recovery status when participants wore FIR-emitting trousers for 2 h immediately after an eccentric exercise bout ($3 \times 30$ maximal eccentric contraction of KE at $60^\circ.s^{-1}$) and over the three consecutive days prior to testing again. When a FIR-emitting long sleeve shirt and trousers were worn between two bouts of exercise (3-h exposure time), similar results were observed on the recovery of neuromuscular function (i.e. MVC, twitch torque and voluntary activation), jump performance, blood CK, perceived pain and recovery [36]. The effects of FIR-emitting stockings following a strenuous exercise (35-min treadmill uphill walking at +10% slope) and worn during the 62-h recovery period (excluding overnight use but used during experimental procedures) was tested in older healthy individuals [37]. Authors did not report significant differences between conditions on resting or exercise (plantar flexion exercise) microvascular perfusion (mBF) or submaximal oxygen consumption ($m\dot{V}O_2$). Possible to likely beneficial effects (as measured by Cohen's effect sizes) were observed on resting mBF and $m\dot{V}O_2$ from 12h to 62h post-exercise, and a possible improvement in exercising $m\dot{V}O_2$ during a concentric plantar flexion exercise (at 10% and 20% baseline-MVC) performed 12h post-exercise [37]. In addition, possible to likely improvements of ankle dorsiflexion and ankle plantar flexion were recorded 26-h and 62-h post-exercise thanks to FIR-emitting garments [37].

## Discussion

The utilisation of FIR-emitting garments is growing in sport, fostered by elite athletes and sport clothing companies. Based on the principles of FIR therapy which has been used for decades in the medical domain, it has been suggested that these innovative clothes could improve exercise performance and recovery in athletes. Since there was no consensus about the effects of FIR-emitting garments worn during and/or after exercise, the aim of this systematic review was to provide a synthesis of current scientific evidence on their potential for enhancing physical performance and/or recovery in healthy people. It is also our research team's viewpoint that further investigations must be oriented at an early stage so that research can serve athletes and practitioners first. Accordingly, a secondary aim of this review was to appraise the level of current evidence and identify areas that require more research.

This review identified eleven studies that investigated the effects of wearing FIR-emitting garments during exercise (n = 6) and during post-exercise recovery (n = 5). Only one of the eleven studies investigated the effect of FIR-emitting garments worn during exercise in ecological conditions during a 10-km running race [34]. Authors reported a possible small effect (ES = -0.23), albeit not significant (p = 0.06), on performance time on the running race thanks to FIR-emitting garments compared to other garment conditions. Possible mechanisms which could explain a performance enhancement include haemodynamic changes, better thermoregulation, or a lower exercise-induced fatigue, as suggested by *in vitro* and animal studies (e.g. [3,7,8,40]). Leung et al. [7] reported that FIR-irradiated isolated amphibian muscles were more resistant to fatigue during exercise and accumulated less metabolic by-products. While this pilot study is not without scientific bias (e.g. initial strength contraction was different between control and irradiated muscles when electro-stimulations were applied to the muscle), it raises the question of the potential effect of FIR on resistance to muscle fatigue. Additional studies are required to investigate the peripheral changes associated with wearing FIR-emitting garments, by investigating for instance the effects on peripheral fatigue (e.g. via the interpolation twitch technique), muscle metabolic changes (e.g. via near-infrared spectroscopy) and perceived muscle effort or pain.

One of the eleven eligible studies also investigated the effects of a FIR-emitting whole-body suit worn during standing postural tests in healthy males, and a handstand stabilization task in expert male gymnasts [32]. In this study, postural control was improved in both non-athletes and expert gymnasts thanks to FIR-emitted garments worn during the task. This was evidenced through lower postural oscillations on a force platform, but the exact physiological effects remain unknown. Further studies investigating muscle activity (e.g. via electromyography techniques) during exercise might help understand the effects of FIR-emitting garments on postural control.

Given the proposed action principle of FIR-emitting garments, it is surprising that very few studies investigated their influence on the body thermal responses (n = 2; [16,30]). During exercise, the increase in skin temperature tended to be lower by wearing FIR-emitting garments compared to a control or placebo garment [16,30]. FIR would improve local microcirculation and peripheral blood flow during exercise [30], which may enhance heat loss [41]. By improving thermoregulation during exercise, FIR-emitting garments could attenuate the alterations in thermal comfort, thermal sensation, and sweating classically recorded during exercise. In a hot environment, FIR-emitting garments might help prevent hyperthermia thanks to their haemodynamic and thermoregulation effects, and could therefore limit the alteration of physical performance [42]. On another hand, wearing FIR-emitting garments could be an interesting strategy to limit evaporative resistance (compared to a control garment) in cold environmental condition, limiting the rapid decrease in body temperature, thermal discomfort, and potentially hypothermia [43]. However, these assumptions are yet to be investigated. Moreover, FIR-emitting socks have been reported to present antiperspirant, bacteriostatic, and cooling properties during long distance running exercise [21]. Another important component of performance in sport which was often neglected in the studies presented previously, is the perceived comfort associated with wearing FIR-emitting garments. For example, we can hypothesise that more comfort during exercise may act positively on perceived exertion and promote the use of FIR-emitting garments in the long term. Further research studies should thus not only consider the physiological effects of wearing FIR-emitting garments, but also their psychological component.

Five studies tested FIR-emitting garments during the post-exercise recovery period with mixed outcomes. In three of these studies [35,38,39] the emphasis was made on the post-exercise overnight period exclusively, whereas in the 2 others FIR-emitting garments were worn in between exercise bouts [36,37], also including sleep time in Zinke et al. [36]. No significant benefits from FIR-emitting garments were found on the recovery of muscle function (i.e. assessed through isometric and dynamic muscle strength, and sprint performance) in all studies, regardless of the exposure time [35,36,38,39]. Small to large Cohen's effect sizes suggested a faster recovery of some indirect markers of muscle damage including DOMS [35,36,39], oxidative stress [39], inflammation [39] and range of motion [37]. However, no significant effect was observed for blood CK elevation among studies [35,36,38]. Interestingly, it seems that the longer the exposure duration to FIR, the greater the benefits on various markers of recovery, and all the more when the exposure is delayed after exercise. Loturco et al. [35] suggested that wearing FIR-emitting garments immediately after intense and/or exercise inducing muscle damage may prompt the development of oedema and leukocyte migration into tissues [44]. In vitro studies demonstrated that irradiation of endothelial cells increased nitric oxide release after 30min [45], resulting in vasodilation and a rise in tissue temperature, thus facilitating neutrophil migration [46,47]. Therefore, it is possible that delaying the application of FIR-emitting garments during the recovery period could downregulate the production of inflammatory mediators that contribute to perceived soreness. Accordingly, the time lapse between the end of exercise and the use of FIR-emitting garments should be investigated or at least

considered in future studies. In addition, the overnight application of FIR-emitting garments may limit external confounding factors (e.g. exposure time, ambient condition, physical activity) and increase compliance from participants, compared to applications during daily activities and/or training sessions [39]. These early results warrant the need for more research to understand whether a dose-response effect may occur in the exposure to FIR-emitting garments and when exposure should take place in athletes.

A secondary aim of his review was to appraise the level of evidence and quality of studies on the utilisation of FIR-emitting garments in sport for improving performance and/or recovery to identify areas for future research. Our review reveals a large variety in the experimental designs, study populations, exercise protocols, and reported variables in published scientific studies, which prevents from drawing firm conclusions on the benefits of FIR-emitting garments in sport. Among the selected studies, the quality of the reporting of methods used was also varied with some PEDro items clearly underachieved. Future studies should therefore target these items to eliminate sources of bias in the results. FIR-emitting garments were also very different among studies, with different designs (i.e. different tightness, and body area covered) and materials (i.e. metallic oxide composition, emissivity), which could likely influence the biological measured outcomes. Both the design and materials embedded in the fabrics could influence the conduction-convection balance and the level of FIR irradiation. For example, we can hypothesise that a tighter FIR-emitting garment could enhance the energy conduction between bioceramic materials and the body's targeted area, and thus potentiate their biological effects. Future studies should therefore clearly define the design of the product under test as it has been poorly described thus far. FIR-emitting garments that embed bioceramic materials (in the form of powders or fibres) directly into the fabrics might also be more effective in providing FIR back to the body compared to bioceramic disks placed onto the garment; the latter solution conferring a lower contact area with the human body. Caution should also be taken when comparing the results obtained with different FIR-emitting techniques such as FIR saunas, FIR ray devices, and FIR-emitting garments. FIR saunas and electrically powered devices have greater irradiance or power density than human body powered garments [2], and thus confer much higher tissue heating capacity. For example, Hausswirth et al. [48] compared the effect of a whole-body FIR-electrically powered device to a whole-body cryostimulation, and passive recovery following a simulated trail run on a treadmill. Authors reported a faster recovery of knee extensors' maximal force production using FIR therapy compared to passive recovery. Because the FIR-electrically powered device not only emitted FIR but also heat (45˚ Celsius measured in the device), it is possible that the hot temperature accounted for all or part of the faster recovery of muscle strength compared to the control recovery strategy.

## Conclusion

The interest for FIR-emitting garments is growing both in the textile industry and sport. By embedding biomaterials within the fabric, these innovative garments have the capacity to interact with the body's physiological functions at rest and during exercise. Garments can absorb body-emitted heat energy, and then re-emit it through radiation (within the FIR wavelength range) back to the body. The main action mechanisms of FIR-emitting garments would be an alteration of the body's thermoregulatory and haemodynamic response while in contact with the skin. Although these effects have been proved useful for many purposes in medicine and physiotherapy, their application in sport is still unclear. This review highlights that FIR-emitting garments might present an interest for optimising physical performance or post-exercise recovery in healthy individuals. However, the low sample sizes, and the variety of protocols and garments tested, prevent from drawing firm conclusion with regards to their

utilisation with athletes. Accordingly, despite the growing presence of these innovative garments on the market, and their numerous potential applications in sport, we recommend keeping a cautious approach. Additional research is required before recommendations can be made to practitioners and athletes. Future research should primarily aim to understand the physiological effects of these garments in athletes at rest, and then test them during exercise and sporting activities.

## Supporting information

**S1 Fig. SR_FIR_PRISMA 2009 checklist.**
(PDF)

## Author Contributions

**Conceptualization:** Bastien Bontemps, Mathieu Gruet, Fabrice Vercruyssen, Julien Louis.

**Data curation:** Bastien Bontemps, Julien Louis.

**Formal analysis:** Bastien Bontemps, Mathieu Gruet, Fabrice Vercruyssen, Julien Louis.

**Investigation:** Bastien Bontemps.

**Methodology:** Bastien Bontemps, Mathieu Gruet, Fabrice Vercruyssen, Julien Louis.

**Project administration:** Bastien Bontemps, Julien Louis.

**Supervision:** Mathieu Gruet, Fabrice Vercruyssen, Julien Louis.

**Writing – original draft:** Bastien Bontemps, Mathieu Gruet, Fabrice Vercruyssen, Julien Louis.

**Writing – review & editing:** Bastien Bontemps, Mathieu Gruet, Fabrice Vercruyssen, Julien Louis.

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
