## [Decision Letter · Decision Letter 0]

1 Apr 2021

PONE-D-21-06115

Utilisation of far infrared-emitting garments for optimising performance and recovery in sport: real potential or new fad? A systematic review

PLOS ONE

Dear Dr. Louis,

Thank you for submitting your manuscript to PLOS ONE. After careful consideration, we feel that it has merit but should be slightly amended before publication, especially by shorten it a bit. Therefore, we invite you to submit a revised version of the manuscript that addresses the points raised during the review process.

We look forward to receiving your revised manuscript.

Kind regards,

Laurent Mourot

Academic Editor

PLOS ONE

Journal Requirements:

Reviewers' comments:

Reviewer's Responses to Questions

**Comments to the Author**

1. Is the manuscript technically sound, and do the data support the conclusions?

Reviewer #1: Yes

Reviewer #2: Yes

2. Has the statistical analysis been performed appropriately and rigorously? 

Reviewer #1: Yes

Reviewer #2: N/A

3. Have the authors made all data underlying the findings in their manuscript fully available?

Reviewer #1: Yes

Reviewer #2: Yes

4. Is the manuscript presented in an intelligible fashion and written in standard English?

Reviewer #1: Yes

Reviewer #2: Yes

5. Review Comments to the Author

Reviewer #1: The present is a systematic review on far infrared-emitting garments for optimizing exercise performance. The review is clearly written and very interesting. I checked data in the literature and find that the review is complete. I also agree with the conclusion that more data are definitely needed on the effects of far infrared-emitting garments on exercise. Most importantly, the type of exercise must be standardized. I have only 2 comments:

a) the discussion is by far too long and can be extensively shortened.

b) at page 7, the authors state that age ranged between 18 and 35 years, but, being one of the subjects who tested this garment, I can tell you that unfortunately the age of participants is in some cases higher.

Reviewer #2: The effects of Far Infrared (FIR) emitting garments are not widely studied in the scientific literature to date. Thus, this review is of major interest in summarizing the effects of FIR emitting garments during exercise and recovery period. This review is well written, the methodology used is consistent and well informed despite the fact that few studies could be included in the review (n = 11). In their conclusion, the authors clearly highlight that FIR emitting garments could have an interest during exercise and recovery phase, however, additional studies must be carried out in particular with more subjects or more varieties of protocols.

6. PLOS authors have the option to publish the peer review history of their article (what does this mean?). If published, this will include your full peer review and any attached files.

Reviewer #1: **Yes: **Piergiuseppe Agostoni

Reviewer #2: **Yes: **G Doucende

---

## [Author Response · Author response to Decision Letter 0]

6 Apr 2021

Dear Editor of PlosOne,

Thank you for giving us the opportunity to revise our manuscript and to the reviewers’ comments. Accordingly, we have corrected the age range of participants included in studies selected in our systematic review and amended the discussion section to make it more succinct. Below is a point-by-point response to reviewers. We hope the article is now suitable for publication in PlosOne. 

Yours sincerely, 

Julien Louis

Reviewer #1: The present is a systematic review on far infrared-emitting garments for optimizing exercise performance. The review is clearly written and very interesting. I checked data in the literature and find that the review is complete. I also agree with the conclusion that more data are definitely needed on the effects of far infrared-emitting garments on exercise. Most importantly, the type of exercise must be standardized. I have only 2 comments:

a) the discussion is by far too long and can be extensively shortened.

b) at page 7, the authors state that age ranged between 18 and 35 years, but, being one of the subjects who tested this garment, I can tell you that unfortunately the age of participants is in some cases higher.

Many thanks for your review and relevant comments. 

a) We have attempted to shorten the discussion, hence making it more succinct and strength to the point. All main paragraphs of the discussion have been amended accordingly, by removing sentences that were either repetitions or too hypothetical (with no sufficient scientific backup). These sections can be found in red in the revised manuscript and correspond to p12 L16-18; p12 L24-26; p13 L23-26; p15 L20-21 in the original manuscript. As a result, the discussion has been reduced by half a page. 

b) thank you for pointing out this, it has been corrected with the full age range (18-57yrs) of participants in selected studies. 

Reviewer #2: The effects of Far Infrared (FIR) emitting garments are not widely studied in the scientific literature to date. Thus, this review is of major interest in summarizing the effects of FIR emitting garments during exercise and recovery period. This review is well written, the methodology used is consistent and well informed despite the fact that few studies could be included in the review (n = 11). In their conclusion, the authors clearly highlight that FIR emitting garments could have an interest during exercise and recovery phase, however, additional studies must be carried out in particular with more subjects or more varieties of protocols.

Many thanks for your positive comments.

---

## [Decision Letter · Decision Letter 1]

23 Apr 2021

Utilisation of far infrared-emitting garments for optimising performance and recovery in sport: real potential or new fad? A systematic review

PONE-D-21-06115R1

Dear Dr. Louis,

We’re pleased to inform you that your manuscript has been judged scientifically suitable for publication and will be formally accepted for publication once it meets all outstanding technical requirements.

Kind regards,

Laurent Mourot

Academic Editor

PLOS ONE

Additional Editor Comments (optional):

Reviewers' comments:

Reviewer's Responses to Questions

**Comments to the Author**

1. If the authors have adequately addressed your comments raised in a previous round of review and you feel that this manuscript is now acceptable for publication, you may indicate that here to bypass the “Comments to the Author” section, enter your conflict of interest statement in the “Confidential to Editor” section, and submit your "Accept" recommendation.

Reviewer #1: All comments have been addressed

2. Is the manuscript technically sound, and do the data support the conclusions?

Reviewer #1: Yes

3. Has the statistical analysis been performed appropriately and rigorously? 

Reviewer #1: Yes

4. Have the authors made all data underlying the findings in their manuscript fully available?

Reviewer #1: Yes

5. Is the manuscript presented in an intelligible fashion and written in standard English?

Reviewer #1: Yes

6. Review Comments to the Author

Reviewer #1: The present is a systematic review on far infrared-emitting garments for optimizing exercise performance. The review is clearly written and very interesting. I checked data in the literature and find that the review is complete. I am satisfied with the revision and have no further comments.

7. PLOS authors have the option to publish the peer review history of their article (what does this mean?). If published, this will include your full peer review and any attached files.

Reviewer #1: **Yes: **Piergiuseppe Agostoni

---

## [Editor Report · Acceptance letter]

27 Apr 2021

PONE-D-21-06115R1 

Utilisation of far infrared-emitting garments for optimising performance and recovery in sport: real potential or new fad? A systematic review 

Dear Dr. Louis:

I'm pleased to inform you that your manuscript has been deemed suitable for publication in PLOS ONE. Congratulations! Your manuscript is now with our production department. 

Kind regards, 

on behalf of

Dr Laurent Mourot 

Academic Editor

PLOS ONE